# Research Progress of *Piriformospora indica* in Improving Plant Growth and Stress Resistance to Plant

**DOI:** 10.3390/jof9100965

**Published:** 2023-09-26

**Authors:** Liang Li, Yu Feng, Fuyan Qi, Ruiying Hao

**Affiliations:** School of Chemical Engineering and Technology, Hebei University of Technology, Tianjin 300130, China; fy15132335384@gmail.com (Y.F.); 13463466257@163.com (F.Q.); haory123@126.com (R.H.)

**Keywords:** *Piriformospora indica*, biotic stress, abiotic stress, root endophytic fungus

## Abstract

*Piriformospora indica* (*Serendipita indica*), a mycorrhizal fungus, has garnered significant attention in recent decades owing to its distinctive capacity to stimulate plant growth and augment plant resilience against environmental stressors. As an axenically cultivable fungus, *P. indica* exhibits a remarkable ability to colonize varieties of plants and promote symbiotic processes by directly influencing nutrient acquisition and hormone metabolism. The interaction of plant and *P. indica* raises hormone production including ethylene (ET), jasmonic acid (JA), gibberellin (GA), salicylic acid (SA), and abscisic acid (ABA), which also promotes root proliferation, facilitating improved nutrient acquisition, and subsequently leading to enhanced plant growth and productivity. Additionally, the plant defense system was employed by *P. indica* colonization and the defense genes associated with oxidation resistance were activated subsequently. This fungus-mediated defense response elicits an elevation in the enzyme activity of antioxidant enzymes, including superoxide dismutase (SOD), peroxidase (POD), and catalase (CAT), and, finally, bolsters plant tolerance. Furthermore, *P. indica* colonization can initiate local and systemic immune responses against fungal and viral plant diseases through signal transduction mechanisms and RNA interference by regulating defense gene expression and sRNA secretion. Consequently, *P. indica* can serve diverse roles such as plant promoter, biofertilizer, bioprotectant, bioregulator, and bioactivator. A comprehensive review of recent literature will facilitate the elucidation of the mechanistic foundations underlying *P. indica*–crop interactions. Such discussions will significantly contribute to an in-depth comprehension of the interaction mechanisms, potential applications, and the consequential effects of *P. indica* on crop protection, enhancement, and sustainable agricultural practices.

## 1. Introduction

Food security has emerged as a looming concern within the context of global climate change, capturing widespread attention. As plants go through the process of growing, they encounter the dual challenge of biotic stresses brought by pathogenic microorganisms and abiotic stresses like drought or floods [1]. These adversities pose threats to plant growth, development, and, ultimately, grain production. Nonetheless, throughout their long evolutionary history, plants have developed the ability to establish mutually beneficial symbiotic associations with beneficial microorganisms present in the rhizosphere. This symbiotic relationship leads to interdependence between plants and microorganisms, enhancing their resilience to external adversities [2]. One such microorganism of interest is *Piriformospora indica*, a root endophyte originally isolated from the rhizosphere of a woody shrub in the Thar Desert of northwestern India [3] and derived its name from the pear-shaped chlamydospore it forms (Figure 1) [4]. Notably, *P. indica* exhibits the capacity to colonize a wide range of plant species, including monocotyledons such as *Triticum* spp. (wheat), *Hordeum vulgare* (barley), *Oryza sativa* (rice), and *Zea mays* (maize), as well as dicotyledons like *Arabidopsis thaliana* [5].

*P. indica* differs from arbuscular mycorrhizae fungi (AMF) in that AMF is usually unculturable, whereas *P. indica* can be cultured on synthetic media without a live host [6]. The colonization of *P. indica* was found to promote host plant growth as well as to enhance stress resistance. It has been found that *P. indica* colonization of passion fruit plants led to a mild defense response that improved both growth and fruit quality [7]. *Centella asiatica* is widely used in medicine and cosmetology, among which the most important active factor is asiaticoside. *P. indica* can be used as a biological inducer to increase asiaticoside concentrations [8]. *P. indica* also promotes earlier flowering, higher biomass, and alterations in secondary metabolites for medicinal plants [9]. The increased biomass, chlorophyll, and potential anticancer agents of *P. indica*-colonized aloe renders aloe more commercially valuable [10]. *P. indica* expresses a phosphate transporter that can promote phosphate uptake by plant roots, providing necessary nutrients for plants [11]. Numerous studies have provided evidence that *P. indica* exerts a positive influence on plant growth and improves plant tolerance to abiotic stress by modulating host enzyme activity, manipulating hormone levels, and upregulating relevant defense genes. The beneficial effects of *P. indica* colonization on plant nutrient absorption and its role in enhancing tolerance to abiotic stresses, including heavy metal, salinity, waterlogging, and drought were summarized here.

## 2. *P. indica* Promotes Nutrient Acquisition in Plant

*P. indica*, an endophytic fungus, plays a crucial role in enhancing nutrient acquisition in plants. When co-cultivated with plant roots, it improves their nutritional status and boosts nutrient absorption. For instance, *P. indica* inoculation has been shown to enhance zinc absorption in plants with low zinc efficiency by increasing the amount and quality of branched roots [12]. Similarly, co-inoculation of *P. indica* with beneficial bacteria such as *Azotobacter chroococcum* WR5 or rhizobium activates nitrogen and phosphorus uptake and metabolism in plants [13,14,15,16]. Nitric oxide (NO) is a pivotal signaling molecule that exerts a significant influence on the interaction between microorganisms and plant roots. Studies have indicated that *P. indica* can induce the generation of NO signals, which influences plant root architecture and nitrogen uptake, ultimately promoting overall plant growth [17]. Moreover, it can promote the growth of *Arabidopsis* and tobacco by stimulating nitrogen accumulation in seedling roots and enhancing the expression of nitrate reductase and starch-degrading enzyme glucan-water dikinase (SEX1) genes in roots [18]. Furthermore, the presence of *P. indica* enhances the expression of phosphate transporter genes (Pt*PT3*, Pt*PT5*, and Pt*PT6*) in the roots of *Poncirus trifoliata* seedlings [19] and *Pinus tabulaeformis* plants [20], highlighting the role of *P. indica* in improving phosphorus uptake and transport.

It is worth noting that the accumulation of phosphate in plants is not solely achieved through the active absorption by plants themselves but also facilitated by the transport capabilities of *P. indica*. It has been proved that *P. indica* inoculation promotes root growth and enhances the yield quantity and quality of crops such as sunflower, rapeseed, and rice by increasing the root’s absorption of essential elements, including nitrogen, phosphorus, and potassium [21,22]. Hence, the colonization of *P. indica* can be considered an indirect means by which plants facilitate nutrient absorption. Moreover, *P. indica*’s colonization enhances nutrient uptake and maintains ion homeostasis by controlling ion accumulation, thereby limiting the transfer of sodium (Na^+^) and potassium (K^+^) ions in plants and influencing gene transcription. Nevertheless, the precise regulatory mechanisms underlying plant nutrient uptake and transformation with *P. indica* colonized remain inadequately elucidated and require further investigation.

## 3. *P. indica* Mediates Plant Tolerance to Abiotic Stress

### 3.1. P. indica Enhances Heavy Metal Tolerance in Plants

Heavy metal pollution in soil environments poses a substantial threat to plant growth, development, and human health. The symbiotic association between the endophytic fungus *P. indica* and plants offers a promising approach to enhance plant traits and remediate heavy metal-contaminated soils. Studies demonstrated that *P. indica* colonization improves arsenic tolerance by reducing hydrogen peroxide accumulation while increasing glutathione and proline levels in arsenic-treated plants and immobilizing arsenic in roots while facilitating iron transport to the leaf tips in rice [23,24]. Furthermore, *P. indica* induces the up-regulation of *glyoxalase I* (*Gly I*) and *glyoxalase II* (*Gly II*) genes, optimizes the redox status of the ascorbic acid-glutathione (AsA-GSH) cycle, and reduces the content of malondialdehyde (MDA) and methylglyoxal (MG), consequently enhancing arsenic tolerance in rice seedlings [24]. *P. indica* also improves the antioxidant defense system and alleviates oxidative stress induced by arsenic, thereby protecting the photosynthetic system of plants from arsenic toxicity [25]. Meanwhile, it increases the proportion of cadmium in the cell wall, decreases its proportion in membrane/organelles and soluble fractions, and significantly enhances peroxidase (POD) activity and glutathione (GSH) content in tobacco [26]. In studies on herbage growth under *P. indica* colonization, cadmium absorption and accumulation were promoted in roots, and the translocation of heavy metals to aboveground was inhibited which reduced cadmium toxicity and improved plant tolerance [27]. Additionally, investigation on the response of *Medicago sativa* to *P. indica* colonization has revealed that *P. indica* improves the antioxidant defense system, alleviates membrane lipid peroxidation, enhances enzymatic activity, and increases *Medicago sativa*’s ability to resist cadmium stress [28]. Proteomic analysis of rice has shown that *P. indica* mitigates cadmium stress by down-regulating key enzymes involved in the sugar alcohol degradation pathway and reducing essential protein modifications mediated by nitric oxide [29]. It has been concluded that *P. indica* can enhance plant tolerance to heavy metals in two aspects. Firstly, it induces the expression of related genes such as *Hmt1*, *Hmt2*, *CAD1*, *PCS*, *1Gsh2*, and Ta*PCSl*, improving the chelating ability of tobacco roots towards heavy metals. Secondly, it activates the expression of the metallothionein gene (*MT2*), disease-course-related protein gene (*PR2*), and chitinase, while increasing the content of antioxidant enzymes, pigments, and proteins in tobacco leaves, strengthening tobacco’s tolerance to heavy metals, including cadmium, chromium, and lead [30]. Researchers hypothesized that P. indica immobilizes heavy metals within its mycelium in the root epidermis, thus reducing the content of heavy metals in rice [31]. Additional research has shown that *P. indica* colonization can also mitigate cadmium toxicity in sunflower [32] and enhance copper tolerance in *Cassia angustifolia* Vahl. [33], and protect proso millet plants from copper-induced oxidative damage [34]. Moreover, *P. indica* application in lead-contaminated arid soils has been found to increase soil microbial respiration and dissolved organic carbon, thus improving the uptake of beneficial minerals and heavy metal tolerance in wheat [35]. The combined use of *P. indica* and nano Fe-oxide significantly enhanced soil adsorption capacity and petroleum hydrocarbon degradation [36]. Furthermore, our previous study on the mycorrhizal complex formed by *Medicago sativa* and *P. indica* has revealed its effectiveness in increasing enzyme activities of urease, sucrose, and fluorescein diacetate (FDA) hydrolase in soil and degrading polycyclic aromatic hydrocarbons in heavy metal-contaminated soils [37,38]. These findings underscore the potential of *P. indica* in phytoremediation and the restoration of heavy metal-contaminated environments.

Based on the research discussed above, we can draw the following conclusions regarding plant tolerance to heavy metals mediated by *P. indica*: (i). The complex formed between *P. indica* and plant roots can effectively enhance soil enzyme activity in rhizospheric and non-rhizospheric soil. (ii). *P. indica* plays a crucial role in capturing heavy metals from the rhizosphere and accumulating them on the root surface. However, it does not mediate their translocation to aboveground plant tissues. (iii). *P. indica* colonization in root increases the antioxidant defense system, including POD, GSH, proline, and plant hormone concentration including IAA, and GA_3_. (iv). The expression of defense-related genes was activated by *P. indica* colonization in the root. 5. Leaf photosynthesis as well as plant yield are promoted due to the colonization of *P. indica*, thereby enhancing the plant’s tolerance to heavy metals. A clear illustration of the interactions and processes regarding the phytoremediation effect of *P. indica*-colonized plants on heavy metal-contaminated soil is shown in Figure 2. These findings emphasize the potential of *P. indica* in combination with plants for enhancing plant tolerance and remediating heavy metal-contaminated soils, thereby contributing to sustainable agriculture practices.

### 3.2. P. indica Mediates Plant Tolerance to Salinity Stress

Salinity stress poses a significant challenge to plant growth and productivity, resulting in reduced photosynthesis, nutrient imbalances, and hindered growth and yield [39,40]. Nonetheless, the colonization of *P. indica* has shown remarkable effects in enhancing plant tolerance to salt stress. *P. indica* colonization in tomatoes has been observed to increase the assimilation rate of CO_2_, leaf water potential, and transpiration rate [41]. The colonization of *P. indica* has also been shown to enhance salt tolerance in *Mexican sage* and maize by boosting the activity of antioxidant enzymes, as well as increasing the content of ascorbate and proline to mitigate oxidative damage in leaves [42,43]. Furthermore, *P. indica* has been reported to enhance salt tolerance in tomato seedlings by regulating the expression of genes such as SOS1 and NHX, maintaining water status, and influencing nutrient levels in plant roots [44]. The reduction in reactive oxygen species (ROS) levels and the enhancement of hormone levels, such as salicylic acid (SA) and gibberellin (GA), in plants colonized by *P. indica* contributes to salinity stress tolerance [15,43,45].

Under salt stress conditions, *P. indica* colonization can enhance plant photosynthesis by transporting Na^+^/K^+^ to aboveground photosynthetic tissues, maintaining K+ homeostasis and osmotic potential [46]. Decreased photosynthesis in cotton under salt stress could be mitigated by *P. indica* inoculation; thus, plant height, stem diameter, and root-shoot ratio were increased [47]. Furthermore, *P. indica* colonization reprogrammed the gene expression and differentially expressed genes were enriched in hormonal interactions, signaling pathways, and cell wall dynamics [48]. The upregulated expression of ion homeostasis genes (e.g., *NHX2* and *SOS1*) under *P. indica* inoculation, together with decreased malondialdehyde and hydrogen peroxide content, alleviates salt stress in rice [49,50]. The transcription levels of genes such as Os*NAC1*, Os*NAC6*, Os*BZIP23*, and Os*DREB2A* are upregulated in rice seedling leaves colonized by P. indica, contributing to enhanced salt tolerance [51]. Moreover, *P. indica* colonization positively regulates the transcript levels of *PMH* + *ATPase*, *SOS1*, and *SOS2*, thereby improving soybean growth and salt tolerance [52]. Epigenetic studies have found that *P. indica*-colonization conferred maize greater salinity tolerance by increasing methylation levels in leaves and changing DNA methylation [53,54]. These findings highlight the ability of *P. indica* to induce systemic salt stress tolerance of plants by altering host physiology, genome, and proteomics. Overall, *P. indica* has emerged as a promising natural regulator for improving plant tolerance under saline conditions.

### 3.3. The Colonization of P. indica Enhances Plant Tolerance to Waterlogging Stress

Excessive water can impact plant growth, leading to root hypoxia and growth inhibition. However, the utilization of *P. indica* as a biological fertilizer has demonstrated beneficial effects in mitigating the adverse impacts of waterlogging stress on plants. It has been found that *P. indica* colonization increases vitamin C, protein, and soluble sugar content in cabbage, rapeseed, and cotton, and decreases nitrate levels to maintain higher plant metabolism under waterlogging [55]. In *P. indica*-inoculated host plants, root vitality and the enzyme activities on respiration and metabolism were significantly improved to withstand the hypoxic injury to plants under waterlogging stress [56]. Notably, the enzyme activity associated with reactive oxygen species (ROS) removal was significantly improved following the incubation of *P. indica* in rapeseed suffering from waterlogging [57]. Similarly, the application of *P. indica* in cotton being waterlogged can significantly promote plant growth and, thus, increase the yield [58]. Moreover, it is shown that *P. indica* colonization can induce stomatal closure and elevate leaf surface temperature in rice under waterlogging stress; and it is further demonstrated that *P. indica* enhanced rice tolerance against water stress by reducing malondialdehyde content; upregulating the activities of antioxidant enzymes, including superoxide dismutase (SOD), peroxidase (POD), and catalase (CAT); reducing the accumulation of ROS; slowing down plant aging [59]. Proteomic analysis indicated that *P. indica* colonization in barley under waterlogging increased the levels of proteins involved in photosynthesis, antioxidant defense systems, and energy transfer [60]. All these findings above collectively confirm that *P. indica* colonization enhances plants’ tolerance to waterlogging stress. However, there is a notable paucity of research on waterlogging and the molecular mechanisms underlying *P. indica*-induced waterlogging tolerance remain underexplored.

### 3.4. P. indica Colonization Enhances Plant Tolerance to Drought Stress

Closure of stomata, leading to reduced photosynthesis activity, is a crucial factor in diminishing plant growth and yield under drought stress. The use of *P. indica* was found to be effective in improving salt stress tolerance in crop plants, consequently opening an innovative and promising application of this fungus in sustainable agriculture, especially in the areas affected by salinity [61]. According to the previous studies, four main aspects have been identified to summarize the role of *P. indica* in mediating plant tolerance to drought stress:Physiological enhancement: *P. indica* colonization has been shown to increase plant-relative water content (RWC) and proline content, contributing to improved plant-drought tolerance [62,63,64,65,66]. The regulating of root morphology and increased total root surface area, volume, and fresh weight were reported in plants under arid conditions as a result of the colonization of *P. indica*, which enhances plant water acquisition in drought conditions to resist environmental stress [57,58,67,68].Antioxidant enzyme activation: *P. indica* colonization improves the activity of antioxidant enzymes, including SOD, POD, and CAT in plants. These enhancements play a pivotal role in preserving the integrity of cellular biomembranes, regulating intracellular osmotic pressure, and mitigating the peroxidation of membrane lipids, as well as the generation of reactive oxygen species (ROS) under drought stress conditions [69,70,71,72].Regulation of drought-related genes: *P. indica* colonization up-regulated the expression of drought-related genes, such as Bn*GG2*, Bn*D11*, Bn*MPK3*, and Bn*PKL* in *Brassica napus* [70], *LEA14*, *TAS14*, *GAI*, and *P5CS* genes in tomato [66], *DREB2A*, *CBL1*, *ANAC072*, and *RD29A* in maize [71]. The regulatory response of these genes comprehensively enhanced the drought tolerance of plants. Furthermore, molecular mechanism analysis using Gene Ontology (GO) and Kyoto Encyclopedia of Genes and Genomes (KEGG) pathways suggested that *P. indica* improves drought tolerance by promoting the genes involved in abscisic acid (ABA), auxin (IAA), salicylic acid (SA), and cytokinin (CTK) biosynthesis pathway [69,73].Impact on leaf photosynthesis: Despite its colonization on roots, *P. indica* can influence leaf photosynthesis, too. For example, *P. indica* has been found to enhance drought tolerance in rice by delaying leaf curl and increasing leaf temperature [73]. Moreover, the colonization of *P. indica* inhibits the decline in photosynthesis rate, as well as the degradation of chlorophyll and thylakoid proteins in Chinese cabbage caused by drought stress [69].

Drought stress negatively affects various aspects of plant physiology, including enzyme structure, photosynthesis, nutrient uptake/transport, and hormonal/nutritional balances. *P. indica* establishes a symbiotic relationship with plants by regulating physiological, biochemical, and molecular processes, thus enhancing drought tolerance in host plants. *P. indica*-mediated plant abiotic stress tolerance including waterlogging, drought, and salinity was concluded in Figure 3.

## 4. *P. indica* Mediates Plant Resistance to Biotic Stress

### 4.1. P. indica Can Improve Plant Resistance to Various Diseases

Throughout their growth and development, plants frequently encounter numerous biological threats, including bacterial infections, insect pests, and viruses. The traditional method to tackle these issues is to spray pesticides, but while effective, it can lead to detrimental effects on the environment and human health due to the presence of chemical residues in plants. An eco-friendly alternative is to use beneficial microorganisms for disease control, promoting sustainable agricultural development [74]. This approach not only enhances crop yield and quality but also reduces the reliance on chemical pesticides, ensuring food and environmental safety [75].

Biological control, using insects, fungi, or bacteria to combat pests and diseases, is a significant strategy in fighting against agricultural threats. It relies on the protection, utilization, and reproduction of dominant natural enemies, as well as the development of sex hormones for pest and disease control [76]. *P. indica*, as an endophytic fungus, has been extensively investigated for its ability to enhance the resistance of various plants against pathogens. For instance, *P. indica* colonization significantly reduces the incidence of wheat yellow rust and improves wheat yield [77]. The colonization of *P. indica* in onion not only enhances onion growth but also activates defense signaling mechanisms, thereby conferring resistance to Stemphylium Leaf Blight through multiple pathways [78]. Inoculation with *P. indica* effectively protects rhododendron seedlings from disease affection caused by *Phytophthora cinnamomi* or *P. plurivora*, such as leaf wilt and rhizome necrosis [79]. Moreover, *P. indica* colonization in bananas restricts the growth of *Fusarium oxysporum* f. sp. cubense race 4 (FocTR4) and reduces the symptoms of Fusarium wilt, while promoting the development of banana lateral roots, thus enhancing their resistance to pathogens [80]. The innate immunity of the wheat colonized by *P. indica* was enhanced and the resistance to Fusarium crown rot finally improved [81].

### 4.2. The Mechanism by Which P. indica Mediates Plant Disease Resistance

#### 4.2.1. Maintenance of Cellular Integrity and Modulation of Defense Enzyme Activity

*P. indica* inoculation has emerged as a promising biological control technology for soil-borne diseases such as crop root rot, stem blight, and root rot. The evaluation of *P. indica*-mediated resistance in tobacco against pathogens such as *Alternaria longipes*, *Colletotrichum gloeosporioides*, *Pythium ultimum*, *Rhizoctonia solani*, *Phytophthora parasitica*, and *Ralstonia solanacearum* showed promising results, disease spots were significantly reduced and pathogen harm to the host were substantially mitigated in the *P. indica*-colonized plants [82]. Our previous research showed that *P. indica* significantly reduced the disease progress on wheat caused by *F. graminearum* and *R. cerealis* in vivo, but did not show any antagonistic effect on *F. graminearum* and *R. cerealis* in vitro [83]. *P. indica* inoculation has been associated with a lower disease index and incidence of Fusarium head blight, attributed to increased ascorbate and glutathione levels, as well as enhanced antioxidant enzyme activity [82,84]. It was also confirmed that *P. indica* colonization in chickpea plants can enhance the antioxidant defense system and improve the resistance of plants to the pathogen *Botrytis cinereal* [85]. Studies have shown that *P. indica* protects barley roots from the loss of antioxidant capacity caused by Fusarium and increases antioxidant enzyme activity by about 35% [86]. In fact, the interaction between *P. indica* and the host can trigger oxidative bursts that affect host signaling pathways. Consistently, the increase in antioxidant enzyme activity was accompanied by a decrease in hydrogen peroxide content in plants, which protects rice from the infection of *Rhizoctonia solani* AG1-IA, consequently enhancing its resistance to sheath blight [87]. Meanwhile, the integrity of the biomembrane system and the stabilization of intracellular osmotic pressure were well maintained under *P. indica* colonization plants. These findings confirmed that *P. indica* can stimulate antioxidant enzymes, scavenge reactive oxygen species (ROS), and trigger defense responses, thereby enhancing plant resistance to pathogens.

#### 4.2.2. Induction of Defense Gene Expression

*P. indica* induces the expression of defense genes in host plants, leading to enhanced disease resistance [88,89]. The expression of disease resistance proteins (DRPs) in cabbage colonized by *P. indica* was significantly increased [69] and PR1 and PR3 were up-regulated in cucumber pre-inoculated with *P. indica* to resist Meloidogyne infection [90]. The colonization of *P. indica* on barley seedlings upregulates specific genes such as leucine-rich repeat receptor kinases (LRR-RKs) and WRKY transcription factors (WRKY-TFs), which confer strong resistance to powdery mildew infection [91]. The expression of PR-1B and PR2 in tobacco colonized by *P. indica* was promoted, resulting in increased resistance to bacterial wilt [92]. Additionally, *P. indica* induces the expression of defense-related genes in rice, improving its resistance to white leaf blight, and in barley, conferring resistance to powdery mildew [93]. The chitinase Si Chi of *P. indica* inhibits pathogens, such as *Magnaporthe oryzae* and *Fusarium moniliforme* infection, and rapidly induces the expression of pathogenesis-related (PR) genes in rice, conferring disease resistance to blast and malignant seedlings [94]. Obviously, the pathogenesis-related genes were employed by *P. indica* colonization to induce plant disease resistance. These findings highlight the efficacy of *P. indica* as a biological control agent, showcasing its ability to enhance plant resistance against various pathogens. By influencing plant gene expression, *P. indica* contributes to the development of a robust defense system in plants.

#### 4.2.3. Modulation of Hormone Metabolism Pathways

*P. indica* colonization can influence the production of plant hormones, including ethylene (ET), jasmonic acid (JA), gibberellin (GA), salicylic acid (SA), and abscisic acid (ABA). These hormones play critical roles in plant defense responses and interactions with pathogens which are relevant to the resistance of biotic stresses. *P. indica* induces IAA biosynthesis and affects JA accumulation to improve the meristem ability of longan root, thereby promoting host plant growth [95]. Furthermore, it also modulates phosphatidic acid (PA) biosynthesis, which can activate PDK1 and lead to OXI1 and AGC2-2 cascades, and then trigger MAPK cascades, promoting the growth of *Arabidopsis* [96]. Metabolomics and gene expression analysis suggest that *P. indica* induces putrescine biosynthesis through a pathway mediated by arginine decarboxylase 1 (ADC1), thus leading to plant growth promotion [18]. Modulations in plant hormone levels may effectively elicit biological responses such as activating defense mechanisms against pathogens. By detecting conserved microbe-associated molecular patterns (MAMPs), plants can synthesize ethylene to resist disease [97]. Differential gene expression was analyzed in *P. indica*-colonized tomato plants infected by Verticillium wilt [98]. Their results demonstrated that *P. indica* induces transcriptome remodeling by altering the expression of jasmonic acid/ethylene (JA/ET) pathway-related genes to promptly and efficiently trigger basal defense responses against pathogen infection. *P. indica* colonization in cotton inhibits the development of Verticillium wilt by influencing the α-linolenic acid metabolic pathway to impact jasmonic acid (JA) biosynthesis [99]. After inoculation with *P. indica*, the levels of JA, salicylic acid (SA), and ET in some plants increased by 1.5–2.0 times, accompanied by the high expression of defense genes, thereby preventing pathogen infection [100,101]. Lately, studies show that microRNA positively responds to *P. indica* colonization and can enhance plant resistance to pathogens by regulating target genes [94,102]. Furthermore, it was found that *P. indica* colonization increased barley yield and upregulated key genes in the gibberellin synthesis pathway [49]. Silencing these gibberellin synthesis-related genes reduced *P. indica* colonization in barley roots, implying a close relationship between the fungus colonization and the gibberellin synthesis pathway. Recently, it has been demonstrated that *P. indica* promotes gibberellin biosynthesis and *Arabidopsis* early flowering by regulating GA biosynthesis gene expression [103]. The mechanism of plant disease resistance mediated by *P. indica* is illustrated in Figure 4. Taking the above together, it can be preliminarily concluded that *P. indica* can activate multiple components of defense signaling, thereby providing durable defense against multiple pathogens.

#### 4.2.4. Suppressing the Translation of Host mRNAs through RNA Interference 

*P. indica*, functioning as an endophytic fungus, exhibits the capability to establish and maintain mutualistic associations with various plant species, offering benefits to both the plants and *P. indica* itself. The mechanism of cross-kingdom RNA interference (ck-RNAi) plays a pivotal role in fostering this mutually beneficial symbiosis. Small RNAs, specifically microRNAs (sRNAs or miRNAs), serve as microbial protein effectors through ck-RNAi-based regulatory mechanisms and can be secreted to inhibit the translation of target mRNAs in their respective hosts. Transcripts encoding proteins involved in circadian clock and flowering regulation, as well as immunity-related factors, have been identified as potential targets of fungal sRNAs within *P. indica*-colonized *Brachypodium distachyon* plants, underscoring the beneficial activity of *P. indica* [104]. The miRNA profiles involved in the interaction between barley and P. indica have been extensively studied, identifying 42 miRNAs through transcriptome sequencing and annotation analysis. These miRNAs are predominantly associated with the target genes that are implicated in cell division, auxin signal reception, and hormone response, thus exerting a pivotal role in the organism’s adaptive response to both biotic and abiotic stressors [105]. Furthermore, the effector molecule PIIN_08944 originating from *P. indica* has been confirmed to interfere with the salicylic acid (SA)-mediated basal immune responses of the host plant [106]. Notably, research has delved into how *P. indica* regulates the levels of NBS-LRR R mRNAs and their corresponding target miRNAs within plant leaves. This regulatory circuit appears to be linked with the protection of *Oncidium orchid* plants against *Erwinia chrysanthemi* infection [101].

In short, multiple studies have demonstrated that *P. indica* boosts the antioxidant capacity of plants, induces the expression of defense-related genes, influences plant hormone content, and modulates microbial defense signaling pathways to overcome pathogenic attacks. Undoubtedly, these factors cooperatively contribute to the defense against pathogen infections. By harnessing the beneficial effects of *P. indica*, plants can strengthen their resistance to microbial pathogens, and this biological control approach reduces the reliance on chemical pesticides, thereby promoting a more sustainable and eco-friendly agricultural system.

## 5. Conclusions and Future Perspectives

*P. indica*, a mycorrhiza-like endophytic fungus, has exhibited its versatility in colonizing various plant species and manipulating plant hormone signals to induce local and systemic resistance to fungal and viral plant diseases. It offers numerous benefits in terms of nutrient absorption, disease resistance, stress tolerance, and growth promotion. The latest research highlighting the ability of *P. indica* to promote plant nutrient uptake and coordinate plant responses to biotic and abiotic stress is referred to in Table 1.

Over the past decade, multiple reports have confirmed the positive impact of *P. indica* on the growth and yield of different plants, including crops, horticultural plants, and medicinal plants. As a promising tool, *P. indica* holds great potential for development and application in increasing plant yield and improving crop tolerance to adverse environments. To further utilize *P. indica*, the following aspects can be considered:

Exploration of key factors promoting growth and stress tolerance: In addition to *P. indica* having been found to contain a phosphorus transporter that enhances phosphorus uptake in plant roots, little is known about the specific growth-promoting and stress-resistant effectors of *P. indica*. The broader mechanisms by which fungal spores or hyphae of *P. indica* deliver nutrients to plant cells mediated plant growth promoting require further investigation. Omics research, such as genomics and transcriptomics, can provide valuable insights into gene function mining and shed light on these mechanisms.

Expansion of colonization studies: *P. indica* is known to establish symbiotic relationships with 12 plant families, but further exploration of its colonization ability to other plant families is needed. Additionally, *P. indica* colonization in plant roots stimulates the innate immune system and triggers hormonal responses, such as jasmonic acid (JA), ethylene (ET), and gibberellin (GA). Understanding the changes in hormone levels and their role in mediating biological stress responses and plant immunity mechanisms is essential. Clear and unified conclusions regarding the response mechanisms of plants to *P. indica* colonization are still lacking.

Increasing grain yield is a primary goal in modern agriculture. However, heavy reliance on chemical control measures has led to environmental pollution and ecological imbalance. Sustainable agricultural development is crucial in the face of worsening environmental conditions. Programs to increase crop yields should be reliable for producers while also being safer for consumers and the environment. With the deterioration of the ecological environment, the sustainable development of agriculture is a hot topic in today’s society, and it is urgent to improve the utilization rate of agricultural resources and economic benefits, thereby developing green agriculture is an inevitable choice. Under the current situation, endophytic fungi *P. indica* can improve crop quality, increase biomass, and improve local and systemic resistance of plants to pathogens, which opens a new idea for green agriculture.

## Figures and Tables

**Figure 1 jof-09-00965-f001:**
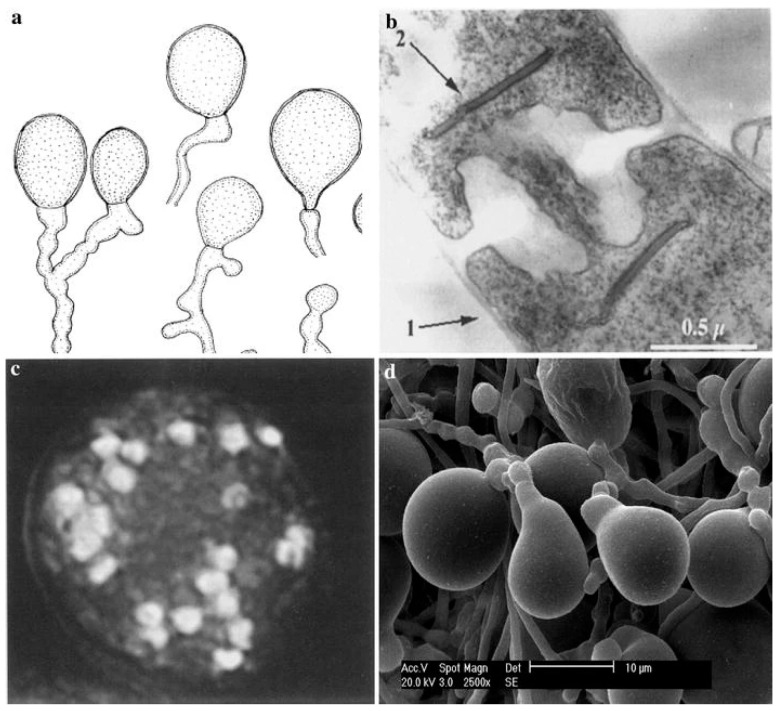
Microstructure observation of *P. indica* (image cited from [4]). (**a**) chlamydospore of *P. indica*; (**b**) Ultrathin sections of *P. indica* mycelium showing dolipore, parenthesomes (arrow 2) and cell wall (arrow 1); (**c**) Nuclei in a chlamydospore; (**d**) Scanning electron microscopy of chlamydospore of *P. indica*.

**Figure 2 jof-09-00965-f002:**
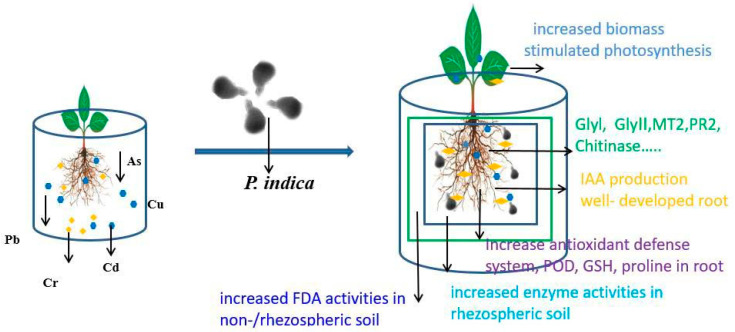
*P. indica* combined with plant contributed to the plant tolerance and soil remediation to heavy metals.

**Figure 3 jof-09-00965-f003:**
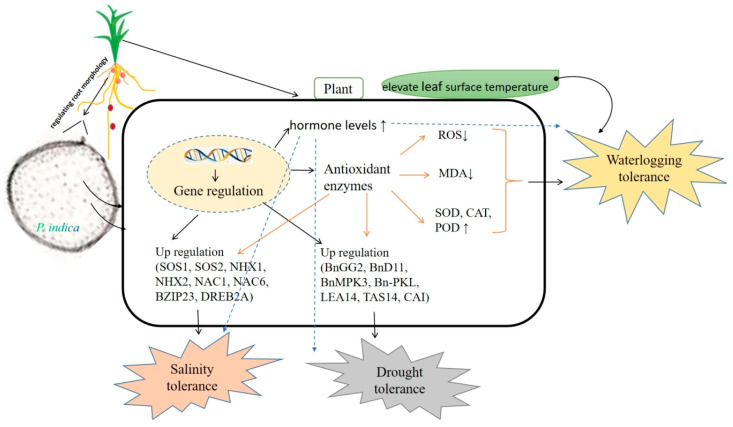
Conclusion of *P. indica*-mediated plant abiotic stress tolerance including waterlogging, drought, and salinity (Color arrows represent antioxidant enzymes-related pathways, dotted arrows represent hormone-related pathways, while up and down arrows indicate up-regulated and down-regulated, respectively).

**Figure 4 jof-09-00965-f004:**
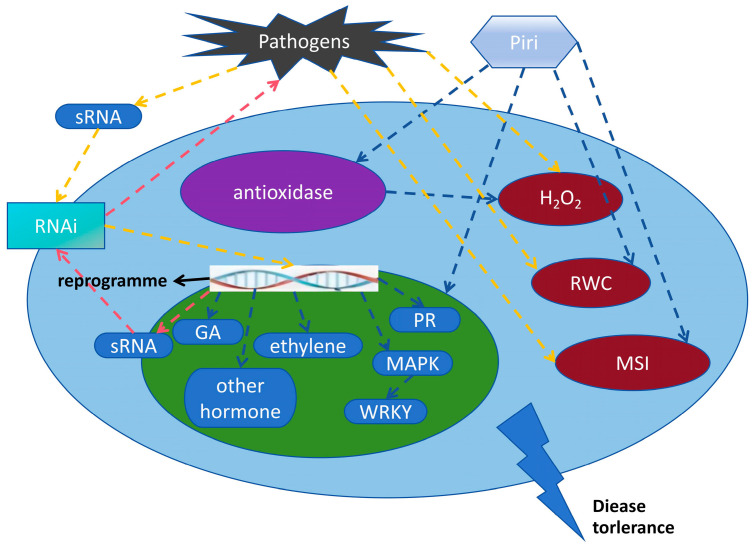
The mechanism of plant disease resistance mediated by *P. indica*.

**Table 1 jof-09-00965-t001:** Studies on the promotion of plant nutrient absorption and the enhancement of abiotic and biotic stress by *P. indica*.

Plant	Nutrient Absorption	Response to Abiotic Stress	Response to Biotic Stress
Wheat [77,87,88,91]	-	Ascorbic acid ↑; Glutathione activity ↑; Antioxidant enzyme activity ↑; relative water content ↑; membrane stability index ↑; POD content ↑; CAT activity ↑	Incidence of Fusarium head blight ↓; Incidence of sharp eyespot and root rot diseases ↓; Incidence of Fusarium crown rot ↓
Chickpea [107]	-	Glutathione activity ↑; CTA activity ↑; SOD activity ↑; H_2_O_2_ content ↓;	*Botrytis cinerea* activity ↓
Tobacco [78,79,80]	N ↑; P ↑; Zn ↑	Pro content ↑; MDA content ↓; SOD activity ↑; POD activity ↑; CAT activity ↑	Incidence of tobacco black shank ↓; Incidence of tobacco bacterial wilt ↓
Onion [81]	-	CTA activity ↑; Phenylalanine ammonia-lyase activity ↑; SOD activity ↑; MDA content ↓	Incidence of Stemphylium leaf blight disease ↓
Banana [108]	N ↑; P ↑; Fe ↓	SOD activity ↑; CAT activity ↑; POD activity ↑; Phenylalanine ammonia-lyase activity ↑; IAA ↑; Pro content ↑	Incidence of Fusarium wilt of banana ↓
Barley [62,86]	-	Glutamine content ↑; Alanine content ↑; Ascorbic acid ↑; Glutathione activity ↑	Incidence of root rot diseases ↓; Incidence of powdery mildew ↓
Maize [43,84,109]	P ↑	ABA ↑; IAA ↑; CTK ↑; SA ↑; CAT activity ↑; Glutathione reductase activity ↑; Glutathione S-transferase activity ↑; SOD activity ↑; ROS ↓	Incidence of root rot ↓; Incidence of seedling blight disease ↓
Rice [18]	-	MDA content ↓; Chlorophyll ↑; Soluble sugar ↑; Soluble protein ↑; Antioxidant enzyme activity ↑; Pro ↑;	Incidence of bacterial blight ↓
*Brassica napus* L [60]	N ↑; P ↑; S ↑; Zn ↑; Mn ↑	CAT activity ↑; MDA content ↓; Membrane permeability ↑; Pro content ↑	Incidence of root rot ↓

“↑” means the plant can upregulate and promote the corresponding item and “↓” means downregulate and inhibit.

## Data Availability

Not applicable.

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
