# Peer review of "Research Progress of Piriformospora indica in Improving Plant Growth and Stress Resistance to Plant"

_jof, 2023, doi:10.3390/jof9100965_

Round 1

Reviewer 1 Report

This is a nice piece of work that is probably worth publishing. However, there are some minor aspects that should be considered before publication.

Heading: Replace "enhancing" by "improving" (of)

 Abstract: 

Mention once the more recently used name for P. indica (Serendipita)

L7: avoid the term “mycorrhizal” in connection with this endophyte

 L12: syntax and 2x “promoting”

 L17: avoid "etc”

 L18: what is "oxidation resistance"?

 L20: not "resistance" genes: authors mean: "defence" genes

L30ff: citations missing

L270 ff: lack of literature references!

L275: fungal toxins? What is the connection to chemical pesticides?

L280ff: shortage required; too repetitive.

 Chapter 4: cite the more general literture on crop protectio by P. indica. Not only the very specific once?

General: Mention some past and recent reviews on P. indica in the introduction! Mention the mile stone papers on P. indica in the Intro.

it needs a complete check of English wording and terms 

Author Response

Thank you for your suggestions. All mentioned mistakes and useful advises have been corrected and adopted. And a few of references was added and several repetitive sections was simplified. Here are some details below:

Heading:

Following your suggestion, i replace “enhancing” by “improving”. And i also replace most “resistance” in the whole article to “tolerance” advised by another reviewer.

Abstract:  

L7: I added the new name Serendipita indica in the place that P. indica first appear. And i deleted the word “endophytic” and just introduced this attribute in the instruction part.

L12: I revised this sentence and replaced the first verb “promotes” to “raises”.

L17/L18: I adjusted the syntax and almost rewrote this line and the the following to make the expressing more accurate. The item “oxidation resistance/resistance to oxidation” refers to the ability of antioxidation, can be used to alloy, plant or even protein.

L20: I have replaced all “resistance genes” to “defense genes” in this paper.

L275: I just deleted “fungal toxins”, i was confused this part with another.

Reviewer 2 Report

The paper is not unnecessarily long and well designated however containing certain repetitions in the phenomenons discussed. These sections, the P. indica effect on nutrient availability, for instance, can be simplified.

Most importantly, the word `resistance` is not used correctly throughout the entire text. Because resistance is mostly for current flow in an electrical circuit in physics. The correct term for plants is `tolerance` here and needs to be changed as suggested in the revised version.

Other than that, there are also some minor points I suggest authors revise accordingly:

L332 LRR receptor kinases and WEKY transcription factors should not be abbreviated

L322-323 please double check the meaning tried to be conveyed here

L15-18 should be reorganized, the connection between the two is not clear

The resolution of definition letters` quality (a, b, and so on) is not enough, and should be increased

Author Response

Thank you for your suggestions. It is so helpful that you mentioned the difference between “resistance” and “tolerance”, and i have finished the replacement. The problem that some sections like nutrient part or biotic part are repetitious have been noticed, and i have adjusted the syntax and deleted the unnecessary words to make it concise and easier to understand. Here are some details below:

L332: I have added the full name of LRR receptor, and WRKY is not abbreviated, it is just the full name of the transcription factors. The statement here have been adjusted, now it would be readable and precise.

L15ff: Another reviewer pointed the same problem, and i have fixed it, reorganizing the whole sentence.

I have changed the figure to a higher resolution, and increased the quality of the letters as best i can do.

Round 2

Reviewer 1 Report

The revised manuscript is much improved!

However, one problem that must be solved is the use of the terms tolerance and resistance. As a single rule:

Tolerance is used in the context of abiotic stress and resistance in the context of biotic stress. This means that in the parts of the manuscript dealing with biotic stress, the term tolerance is used incorrectly. And the title is also incorrect.

The manuscript still contains many spelling errors.

Since there are many reviews on P. indica, an important strategy to make the manuscript more interesting for researchers in the field would be to briefly mention and explain the work on the function of sRNA in the symbiosis of plants with Piriformospora, since this is an emerging field of symbiosis research (see: Secic et al. 2021: PMCID: PMC8385953, DOI: 10.1186/s12915-021-01104-2.

writing errors throughout!

Author Response

Thank you once again for your thoughtful and detailed suggestions. The rule you provided for distinguishing between "resistance" and "tolerance" has been immensely helpful in resolving a long-standing confusion for me. I have made the necessary changes to the usage of these two words in my paper based on the rule and my improved understanding.

In terms of addressing the writing errors, I've put forth my best efforts to revise them, and I've also utilized the Grammarly service to assist in error correction. As a result, I believe the overall quality of the manuscript has improved, although there may still be a few remaining errors that need attention.

Furthermore, in accordance with your advice to enhance the manuscript's overall quality, I have incorporated a small section on sRNA, as indicated in section 4.2.4, and also mentioned it in the abstract. I am hopeful that these additions will contribute to making the manuscript more engaging and distinctive.